# New prolonged opioid consumption after major surgery in Sweden: a population-based retrospective cohort study

Felix C B Lindeberg ![ORCID] ,[1,2] Max Bell,[1,2] Emma Larsson,[1,2] Linn Hallqvist[1,2]

[1]Department of Anaesthesia and Intensive Care Medicine, Karolinska University Hospital, Stockholm, Sweden
[2]Department of Physiology and Pharmacology, Karolinska Institute, Stockholm, Sweden

**Correspondence to**
Felix C B Lindeberg;
felix.lindeberg@stud.ki.se

## ABSTRACT

**Objective** Given that long-term opioid usage is an important problem worldwide and postsurgical pain is a common indication for opioid prescription, our primary objective was to describe the frequency of new prolonged opioid consumption after major surgery in Sweden and, second, to evaluate potential associated risk factors.

**Design** Cohort study including data from 1 January 2007 to 31 December 2014. Data regarding surgical procedures, baseline characteristics and outcomes was retrieved from the Orbit surgical planning system, the Swedish national patient register and the Swedish cause of death register.

**Setting** Observational multicentre cohort study with data from 23 Swedish hospitals.

**Participants** We included 216 877 patients aged ≥18 years, undergoing non-cardiac surgery, not exposed to opioids 180 days before and alive 12 months after surgery.

**Primary and secondary outcome measures** The primary endpoint was collection of at least three opioid prescriptions during the first postoperative year; within 90 days, day 91–180 and 181–365 after surgery in a previously opioid-naïve patient. Second, multivariable logistic regression analysis was conducted to explore potential risk factors associated with prolonged opioid use.

**Results** Of the 216 877 patients identified to undergo analysis, 15 081 (7.0%) developed new prolonged opioid consumption. Several risk factors were identified. Having a history of psychiatric disease was identified as the strongest risk factor (adjusted odds ratio: 1.94; 95% CI: 1.87 to 2.00).

**Conclusion** In a large Swedish cohort of surgical patients, 7% developed new prolonged opioid consumption after major surgery. Our data on susceptible patients could help clinicians reduce the number of prolonged opioid users by adapting their analgesic and preventative strategies.

## INTRODUCTION

The analgesic effect of opioids is a cornerstone of modern medicine but also fraught with side effects ranging from constipation and nausea to substance abuse and death.[1] More than 300 million people undergo surgery in the world every year and postsurgical pain is a common indication for opioid treatment.[2 3] Many patients are exposed to opioids for the first time due to surgery, and it has

---

### STRENGTHS AND LIMITATIONS OF THIS STUDY

⇒ A nationwide observational cohort study investigating opioid prescription after major surgery on a large Swedish population.
⇒ The study cohort includes adults of all ages, exposed to various major surgeries, and data is collected from Swedish health registers which offers robustness and great coverage.
⇒ The lack of consensus in defining prolonged opioid consumption is a weakness of our study and others in the field.
⇒ This study does not portray patient compliance to their prescription as dispensing of opioid prescriptions is a proxy for opioid consumption.
⇒ Lack of long-term follow-up, after the first postoperative year.

---

been suggested that postsurgical long-term opioid use should be referred to as a surgical complication.[4] In addition, opioids are often overprescribed after surgery and many patients report taking opioids for a different indication than for which the opioids were initially prescribed for.[3 5] A previous study has indicated that patients who are prescribed opioids 7 days after surgery are 50% more likely to collect an opioid prescription 1 year after surgery.[6] This is interesting since the proportion of patients receiving an opioid prescription at 7 and 30 days after surgery differs enormously between countries.[7]

Treating postsurgical pain with opioids poses a challenge for physicians as two competing interests need to be addressed: managing acute pain and minimising the risk of prolonged opioid use. A recent study showed that a high percentage of patients with a hip fracture, opioid naive before surgery, were still using opioids several months after their surgical procedure.[8] Furthermore, risk factors for prolonged opioid use have been identified including type of surgery, comorbidities and use of specific drugs ahead of

surgery, for example, antidepressants and benzodiazepines.[8–10] Although some data exists, preoperative risk assessment might stand to gain from more data on risk factors of prolonged opioid use.

Therefore, the primary aim of the study was to describe the frequency of new prolonged opioid consumption after major surgery in Sweden. Data from a Swedish cohort would add insight to the field as most of similar studies have been conducted on North American cohorts. Second, we sought to evaluate the impact of patient factors (age, sex, medical history), surgical factors (procedure, medical treatment) and healthcare system-related factors (regional differences and time trends). Based on earlier studies from other countries, we hypothesised that differences associated with both patient and surgical characteristics exist.

## MATERIALS AND METHODS

This was an observational multicentre cohort study using data from 23 Swedish hospitals at university, county and district level, covering approximately 40% of the Swedish

perioperative care. Included hospitals are located in different legislative regions and dispersed geographically throughout the country which makes the data generalisable to the whole Swedish population. Patients aged ≥18 years, undergoing surgery from 1 January 2007 to 31 December 2014 were included. Exclusion criteria were: collection of an opioid prescription within 180 days ahead of surgery, not being alive 12 months after surgery, undergoing cardiac, obstetric, ambulatory, minor (online supplemental table 1) or multiple surgeries as well as missing a valid surgery code in Orbit or a corresponding surgery code in national patient register (NPR). Additionally, patients identified from hospitals with a high proportion of missing American Society of Anaesthesiologists (ASA) physical status classification were also excluded (figure 1). The opioid-naive period of 180 days aims to exclude patients with a persistent opioid consumption while still including the patients where surgery may act as the trigger for prolonged opioid consumption.

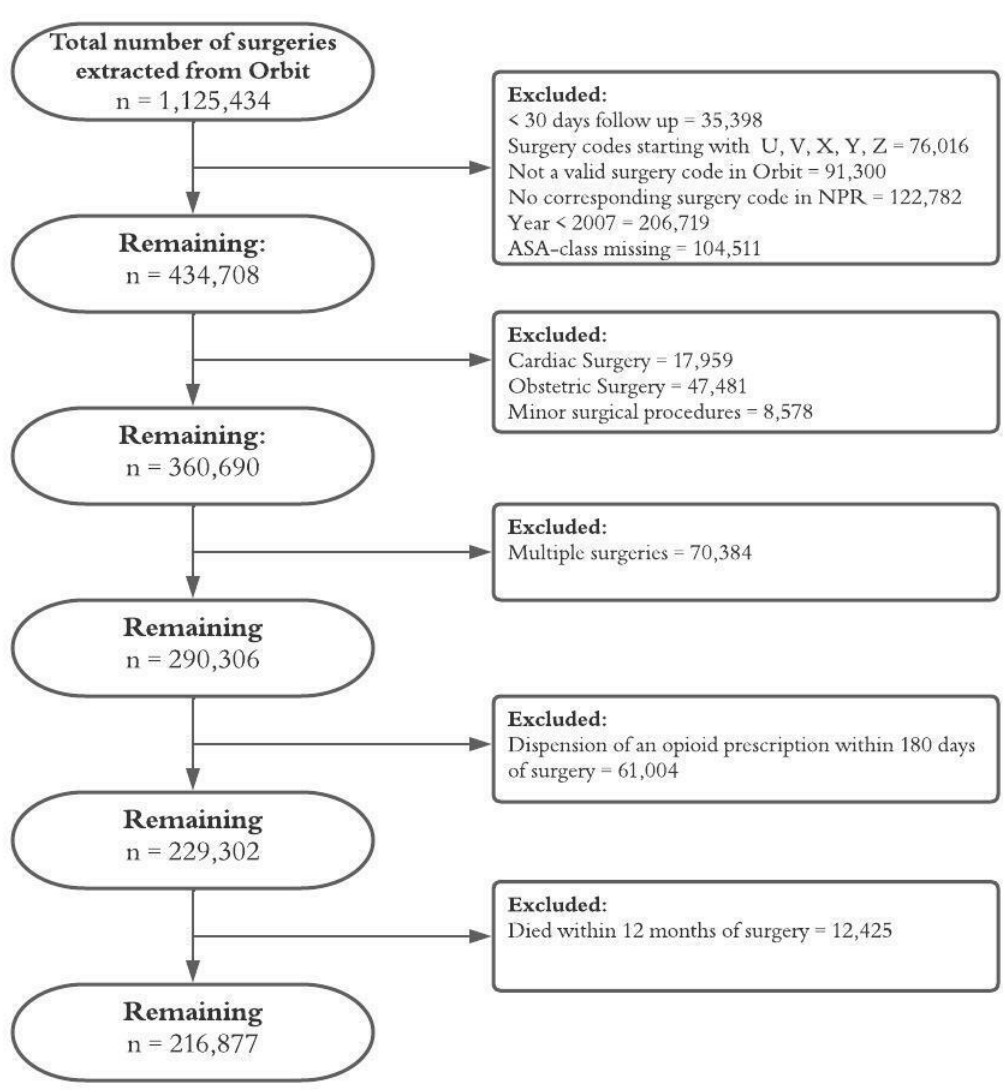

**Figure 1** Participant flowchart.

## Data sources

### Orbit—surgical planning system

Orbit is a surgical planning system used by Swedish hospitals and provides information about patient characteristics, ASA physical status classification, date, type and duration of surgery and anaesthesia, all linked to the Swedish personal identification number (PIN). Surgery cannot be performed without admitting a patient to orbit. The system was used to identify the study cohort.

### Health registers

Data extracted from Orbit were linked to the NPR using the unique Swedish PIN assigned at birth or immigration.[11] The NPR provides information of all hospital admissions since the year of 1987 and outpatient care since 2001.[12] The Swedish prescribed drug register contains information about all dispensed and collected drug prescriptions in Sweden since July 2005.[13] In the Swedish cause of death register, virtually all (> 99%) deaths are registered since 1952 based on the current ICD classification.[14]

### Data collection

Data was retrieved from the above-mentioned registers. Data collection included information about patient characteristics and medical history such as age, sex, ASA physical status classification, hospital diagnoses and collected drug prescriptions. This was made possible by backtracking the NPR and the Swedish prescribed drug register for 5 years before surgery and 1 year after surgery. Data from the Swedish cause of death register was collected for 1 year after surgery. Perioperative characteristics included date, type and duration of surgery and was extracted from Orbit. Surgical procedures were clustered into 12 subtypes based on surgical codes used in the Nordic countries (NOMESCO). Subtypes included gastrointestinal; endocrine; ophthalmic; ear, nose and throat; thoracic; neuro; breast; urologic; gynaecologic; orthopaedic; vascular and dermatologic surgery.

### Primary endpoint

The primary endpoint was new prolonged opioid use defined as collection of at least three opioid prescriptions during the first postoperative year; within 90 days, day 91–180 and 181–365 after surgery in a previously opioid-naive patient. Being opioid naive was defined as not collecting an opioid prescription within 180 days prior to surgery.

### Secondary aims

Secondary aims were to evaluate the impact of patient, surgical and healthcare system-related risk factors for prolonged opioid prescription in the opioid-naive patient. Patient risk factors included age, sex, as well as somatic and psychiatric comorbidities, surgical risk factors include type of surgery and medical treatment, whereas healthcare-related risk factors are differences associated with four different regions in Sweden and the year of surgery.

## Statistical analysis

The data was analysed during the autumn of 2022. The study cohort was divided into two groups based on meeting the primary outcome or not. Categorical variables within each of these groups are presented as percentages of the total population and continuous data as medians with IQRs that represent 25th–75th percentiles. Continuous variables were analysed using the Mann-Whitney U test and categorical variables were compared with the $\chi^2$ test generating p values to look for statistical differences between the groups. P values <0.05 were considered significant and statistical tests were two-sided. Logistic regression analysis was conducted to investigate the relationships between potential preoperative and perioperative, patient-related and surgery-related risk factors and the associated effect on our primary endpoint. Variables with univariate significance, and considered clinically relevant, were explored in the multivariate analysis to explore our secondary aims; results showed in online supplemental table 2. Hence, crude and adjusted ORs (aORs), presented with 95% CIs, were calculated to see how the risk estimates changed when adjusting for potential confounders. Adjustments for age, sex, comorbidities, year of surgery and type of surgery (including cancer surgery) were performed in the final model. Patients that died within 1 year of surgery were excluded due to competing risk. A sensitivity analysis including these patients was conducted which had little effect on the results. A history of psychiatric disease was significantly associated with elevated risk of prolonged opioid consumption, and we hypothesised that the impact of this risk factor might be different depending on age. Therefore, a restriction analysis was performed, using the adjusted model with exclusion of patients >70 years, in the upper fourth quartile (n=67 132). Data was analysed by STATA V.14.2 (Stata Corp).

## Patient and public involvement

It was not possible to involve patients or the public in the design, or conduct, or reporting, or dissemination plans of our research.

## RESULTS

### Patient characteristics

patients were identified to undergo analysis (figure 1). Sixty-one thousand four patients were excluded due to collection of an opioid prescription within 180 days of surgery and 12 425 patients were excluded because they died within 12 months of surgery (figure 1). The population consisted of 118 698 (55%) women and had a median (IQR) age of 62 (45–72) years (table 1). Twenty one thousand one hundred sixty-nine (10%) of patients had a history of psychiatric disease and 78 743 (36%) had collected a prescription for psychiatric medication up to 5 years before surgery (table 1); 152 204 (70%) of the conducted surgeries were elective and orthopaedic surgery was the most common procedure (table 1).

**Table 1** Baseline characteristics of surgical cohort at inclusion (n=216877) and proportion of new prolonged opioid consumption

| | | New prolonged opioid consumption | | |
| | Total n=216 877 (%) | No n=201 796 (93) | Yes n=15 081 (7.0) | P value*† |
|---|---|---|---|---|
| Age—median years (IQR) | 62 (45–72) | 61 (44–72) | 66 (52–76) | <0.001 |
| Women—no (%)‡ | 118 698 (55) | 110 173 (55) | 8525 (57) | <0.001 |
| Age range in years—no (%)‡ | | | | <0.001 |
| 18–29 | 22 996 (11) | 22 118 (11) | 878 (5.8) | |
| 30–39 | 18 509 (8.5) | 17 593 (8.7) | 916 (6.1) | |
| 40–49 | 25 634 (12) | 24 171 (12) | 1 463 (9.7) | |
| 50–59 | 31 973 (15) | 29 732 (15) | 2 241 (15) | |
| 60–69 | 49 472 (23) | 45 894 (23) | 3 578 (24) | |
| 70–79 | 41 394 (19) | 38 989 (19) | 3 314 (22) | |
| >80 | 26 899 (12) | 24 208 (12) | 2 691 (18) | |
| Preoperative data—no (%)‡ | | | | |
| ASA classification | | | | <0.001 |
| ASA1 | 74 067 (34) | 70 947 (35) | 3 120 (21) | |
| ASA2 | 96 381 (44) | 89 173 (44) | 7 208 (48) | |
| ASA3 | 43 853 (20) | 39 406 (20) | 4 447 (29) | |
| ASA4 | 2 576 (1.2) | 2 270 (1.1) | 306 (2.0) | |
| Charlson comorbidity index | | | | <0.001 |
| 0 | 125 868 (58) | 118 447 (59) | 7 421 (49) | |
| 1 | 18 408 (8.5) | 16 684 (8.3) | 1724 (11) | |
| >2 | 72 601 (33) | 66 319 (33) | 10 751 (41) | |
| Heart disease§ | 51 693 (24) | 46 904 (23) | 4 796 (32) | <0.001 |
| Lung disease¶ | 10 457 (4.8) | 9 259 (4.6) | 1 198 (7.9) | <0.001 |
| Renal disease** | 6 079 (2.8) | 5 525 (2.7) | 554 (3.7) | <0.001 |
| Diabetes mellitus | 19 630 (9.0) | 17 694 (8.8) | 1 936 (13) | <0.001 |
| Peripheral vascular disease†† | 6 837 (3.2) | 6 182 (3.1) | 655 (4.3) | <0.001 |
| Cerebrovascular disease‡‡ | 8 323 (3.8) | 7 496 (3.7) | 827 (5.5) | <0.001 |
| Cognitive disease§§ | 3 908 (1.8) | 3 375 (1.7) | 533 (3.5) | <0.001 |
| Substance abuse disease¶¶ | 6 191 (2.9) | 5 413 (2.7) | 788 (5.2) | <0.001 |
| Personality disorder, schizophrenia*** | 5 795 (2.7) | 5 310 (2.6) | 485 (2.6) | <0.001 |
| Affective disorders††† | 8 435 (3.9) | 7 536 (3.7) | 899 (6.0) | <0.001 |
| Anxiety disorders‡‡‡ | 8 774 (4.1) | 7 867 (3.9) | 907 (6.0) | <0.001 |
| History of psychiatric disease§§§ | 21 169 (10) | 19 056 (9.4) | 2 113 (14) | <0.001 |
| Preoperative medication—no (%)‡**** | | | | |
| Psychiatric medication¶¶¶ | 78 743 (36) | 70 692 (35) | 8 051 (53) | <0.001 |
| Neuroleptics | 7 007 (3.2) | 6 211 (3.1) | 796 (5.3) | <0001 |
| Benzodiazepines | 35 721 (16) | 31 853 (16) | 3 868 (26) | <0001 |
| Hypnotics and sedatives | 48 117 (22) | 42 660 (21) | 5 457 (36) | <0001 |
| Antidepressants | 41 008 (19) | 36 549 (18) | 4 459 (30) | <0001 |
| Psychostimulants | 1 431 (0.7) | 1 270 (0.6) | 161 (1.1) | <0001 |
| Anti-dementia drugs | 2 171 (1.0) | 1 867 (0.9) | 304 (2.0) | <0001 |
| Type of surgery—no (%)‡ | | | | <0.001 |
| Acute surgery†††† | 64 673 (30) | 59 259 (29) | 5 414 (36) | |
| Elective surgery | 152 204 (70) | 142 537 (71) | 9 667 (64) | |
| Cancer surgery | 43 881 (20) | 40 613 (20) | 3 268 (22) | <0.001 |
| Time of surgery—min (IQR) | 82 (50–129) | 82 (50–128) | 88 (55–138) | <0001 |

Continued

**Table 1** Continued

| | Total n=216877 (%) | New prolonged opioid consumption | | P value*† |
|---|---|---|---|---|
| | | No n=201 796 (93) | Yes n=15 081 (7.0) | |
| Procedure—no (%)‡ | | | | <0.001 |
| Ophthalmic surgery | 3 228 (1.5) | 3 167 (1.6) | 611 (0.4) | |
| Neurosurgery | 14 052 (6.5) | 13 019 (6.5) | 1 033 (6.9) | |
| Endocrine surgery | 7 573 (3.5) | 7 429 (3.7) | 144 (1.0) | |
| Ear, nose and throat surgery | 6 839 (3.2) | 6 666 (3.3) | 173 (1.2) | |
| Oral and maxillofacial surgery | 8 729 (4.0) | 8 331 (4.1) | 398 (2.6) | |
| Pulmonary surgery | 2 244 (1.0) | 2 023 (1.0) | 221 (1.2) | |
| Breast surgery | 12 635 (5.8) | 12 104 (6.0) | 531 (3.5) | |
| Gastrointestinal surgery | 43 112 (20) | 40 521 (20) | 2 591 (17) | |
| Urologic surgery | 23 642 (11) | 22 509 (11) | 1 133 (7.5) | |
| Gynaecologic surgery | 18 94 (8.8) | 18 412 (9.1) | 582 (3.9) | |
| Orthopaedic surgery | 61 360 (28) | 54 003 (27) | 7 357 (49) | |
| Vascular surgery | 8 450 (3.9) | 7 965 (4.0) | 485 (3.2) | |
| Dermatologic surgery | 6 019 (2.8) | 5 647 (2.8) | 372 (2.5) | |
| Opioid type at discharge—no (%)‡ | | | | |
| Opiates, for example, morphine | 57 015 (26) | 48 882 (24) | 8 133 (54) | <0.001 |
| Fentanyl and Ketogan | 1 179 (0.5) | 922 (0.5) | 257 (1.7) | <0.001 |
| Methadone | 7 675 (3.5) | 6 747 (3.3) | 928 (6.2) | <0.001 |
| Buprenorphine, for example, Suboxone | 139 (0.1) | 86 (0.04) | 53 (0.4) | <0.001 |
| Opioids with antispasmodics, for example, Targiniq | 317 (0.2) | 269 (0.1) | 48 (0.3) | <0.001 |
| Other opioids, for example, tramadol | 18 430 (8.5) | 15 964 (7.9) | 2 466 (16) | <0.001 |
| Year of surgery—year (%)‡ | | | | <0.001 |
| 2007–2008 | 42 195 (20) | 39 192 (19) | 3 003 (20) | |
| 2009–2010 | 46 963 (22) | 43 705 (22) | 3 258 (22) | |
| 2011–2012 | 60 213 (28) | 56 067 (28) | 4 146 (27) | |
| 2013–2014 | 67 506 (31) | 62 832 (31) | 4 674 (31) | |

*Pearson's $\chi^2$ test.
†Mann-Whitney U test.
‡Percentages calculated in relation to vertical study cohort.
§Chronic ischemic heart disease, angina pectoris, hypertensive disease, cardiac arrest, heart failure, valve disease, pulmonary heart disease, cardiomyopathy, conduction disorders/cardiac arrhythmias, cardiac arrest, diseases of arteries, arterioles and capillaries.
¶Pneumonia, COPD.
**Acute renal failure/unspecified renal failure, chronic renal failure, other renal disease.
††Atherosclerosis, aortic aneurysm and dissection, other aneurysm, other peripheral vascular diseases, arterial embolism and thrombosis, atheroembolism, septic arterial embolism, other disorders of arteries and arterioles, diseases of capillaries, disorders of arteries, arterioles and capillaries in diseases classified elsewhere.
‡‡Subarachnoid haemorrhage, intracerebral haemorrhage, other non-traumatic intracranial haemorrhage, cerebral infarction, acute cerebrovascular disease without cerebral infarction, vascular occlusion without cerebral infarction, other cerebrovascular diseases, cerebrovascular disorders in diseases classified elsewhere, sequelae of cerebrovascular disease.
§§Alzheimer's disease, vascular dementia, other dementia, unspecified dementia, non-alcoholic amnesia, non-alcoholic delirium, mental disorders due to known physiological condition, personality and behavioral disorders due to known physiological condition, mental disorder due to unknown somatic or organic disorder.
¶¶Mental and behavioural disorders due to psychoactive substance use.
***Schizophrenia, schizotypal disorder, persistent delusional disorders, acute and transient psychotic disorders, induced delusional disorder, schizoaffective disorders, other non-organic psychotic disorders, unspecified non-organic psychosis.
†††Manic episode, bipolar affective disorder, depressive episode, recurrent depressive disorder, persistent mood disorders, other affective mood disorders, unspecified affective mood disorders.
‡‡‡Phobic anxiety disorders, other anxiety disorders, obsessive-compulsive disorder, reaction to severe stress and adjustment disorders, dissociative disorders, somatoform disorders, other neurotic disorders.
§§§Cognitive disease, substance abuse disorder, personality disorder/schizophrenia, affective disorder, anxiety disorder, psychiatric medication within 5 years of surgery.
¶¶¶Collection of ≥1 of the listed psychiatric medications (neuroleptics, benzodiazepines, hypnotics and sedatives, antidepressants, antipsychotics, anti-dementia drugs) within 5 years of surgery.
****Collection of prescription within 5 years of surgery.
††††Acute surgery is defined as procedures that are admitted as acute surgeries. The level of acuteness can differ between right away, 6 hours and 72 hours depending on type of procedure.
ASA, American Society of Anaesthesiologists.

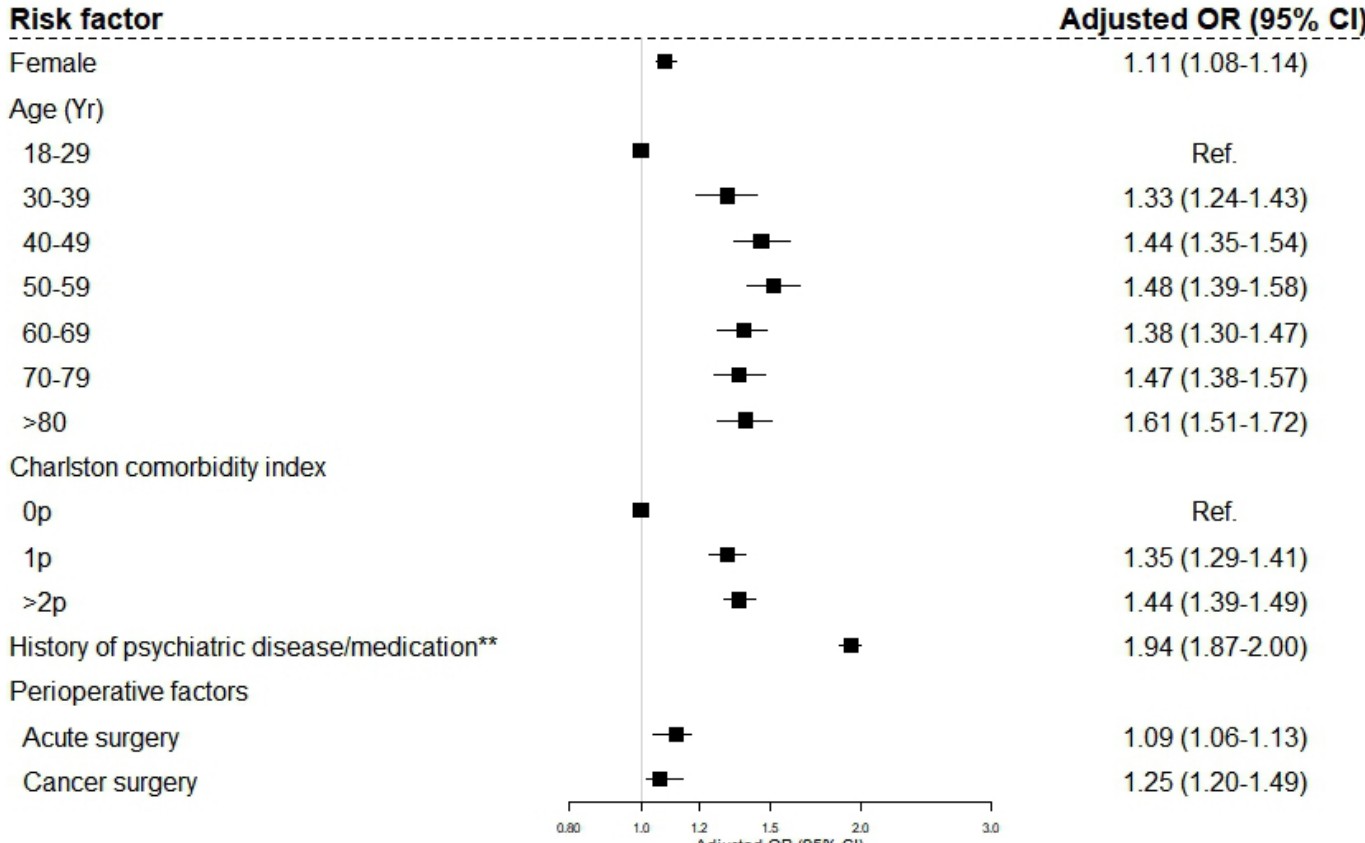

**Figure 2** Risk factors for developing prolonged opioid use after surgery among patients with no opioid use 180 days before surgery. ORs adjusted for sex, age, comorbidities (Charlson comorbidity index) and surgical procedure. \*\*A diagnosis of ≥ 1 of the psychiatric diseases listed (table 1) and/or a collection of ≥ 1 prescriptions of the listed psychiatric medications (table 1) within 5 years of surgery.

### Primary endpoint

A total number of 15 081 patients (7.0 %), opioid naive before surgery, collected at least three opioid prescriptions during the first postoperative year; within 90 days, day 91–180 and 181–365 after surgery (table 1).

### Multivariable analysis of risk factors associated with prolonged opioid use after surgery

Estimated aORs are presented in figure 2. ORs were adjusted for sex, age, comorbidities, year of surgery and type of surgery. Postoperative prolonged opioid use was associated with female sex (aOR: 1.08; 95% CI: 1.05 to 1.12). In terms of the age, there was no striking trend, and the highest risk was found for patients between 50 and 59 years of age (aOR: 1.52; 95% CI: 1.40 to 1.65). Compared with a Charlson comorbidity index (CCI) of 0 p, the risk increased with a higher index. CCI of 1 p had an aOR of 1.31 (95% CI: 1.24 to 1.39) and CCI ≥2 had an aOR of 1.36 (95% CI: 1.30 to 1.43). Having a history of psychiatric disease and/or psychiatric medication use yielded an aOR of 1.94 (95% CI: 1.87 to 2.00). Estimated aORs for ASA class, individual comorbidities and preoperative medications are presented in online supplemental table 2.

Regarding perioperative factors, acute surgery had an aOR at 1.12 (95% CI: 1.04 to 1.17) compared with elective surgery and cancer surgery had an aOR of 1.06 (95% CI: 1.02 to 1.14).

### Sensitivity analysis

Several sensitivity analyses were conducted. The results remained similar when including patients that died within 12 months of surgery in the analysis. No significant time trends or regional differences were identified (online supplemental table 3). As history of psychiatric disease was associated with elevated risk of prolonged opioid consumption, a restriction analysis was performed, using the adjusted model with exclusion of patients >70 years, in the upper fourth quartile (n=67 132). This yielded a higher risk estimate regarding history of psychiatric disease (aOR: 2.07; 95% CI: 1.99 to 2.17), indicating that the impact of a psychiatric disease on risk of prolonged opioid consumption might be higher in younger patients, detailed in online supplemental table 4.

### DISCUSSION

In this large nation-wide retrospective perioperative cohort study, 7% developed new prolonged opioid use.

Factors associated with a higher risk of prolonged opioid use were female sex, more comorbidities and acute surgery. Additionally, having a history of psychiatric disease was identified as the strongest risk factor.

## Strengths

This is a retrospective observational cohort study investigating opioid prescription after surgery on a large Swedish population. To our knowledge, the number of included patients outnumbers many other similar studies on an international level. Utilising the Orbit surgical planning system, this study was able to include patients from different regions all around Sweden covering approximately 40% of Swedish perioperative care. Given inclusion of hospitals from different regions and levels, we believe the results to be representative of the remaining ~60% of Swedish care and hence, generalisable in other Swedish settings. Furthermore, the robustness and great coverage of Swedish health registers is also a strength of this study.[12] Our cohort included adults of all ages being exposed to a great variety of surgeries. Despite not having information about socioeconomic status, our data should represent the actual healthcare burden of the Swedish population. The public healthcare system with services being provided according to patient need limits the selection bias that healthcare systems with various private insurances may generate.

## Limitations

The most important weakness of this study and others in the field is the lack of consensus in defining prolonged opioid consumption. This complicates comparing results in between studies. Furthermore, there is also no consensus for how long patients should be opioid free before surgery.

Some other limitations of our study merit emphasis. First, our study was observational and therefore does not prove that our identified risk factors portray a causal relationship. Moreover, since a large part of our data was extracted from various Swedish registers, misclassification bias could also have played a part in the reporting of data. However, the potential misclassification bias should be non-differential in nature and hence not push the results in one direction or the other. The relatively old data is a potentially limiting factor. But, to our knowledge, there has been no major changes in Swedish postoperative pain management or legislation affecting opioid prescription patterns in recent years. This notion is also supported by the absence of differences over time in our study which makes us believe more recent data would yield similar results. Another important aspect is the fact that we are measuring dispensing of opioid prescriptions as a proxy for opioid consumption. This study does not portray patient compliance to their prescription. However, regardless of actual opioid consumption dispensing of prescriptions are problematic in and of themselves as it increases the number of opioids circulating in communities. Undeniably, we cannot account for surreptitiously

opioid usage either and lack data of in-hospital drug use, something that could influence postoperative opioid usage.

## Postoperative prolonged opioid use

Compared with others, this study found a higher occurrence of new prolonged opioid use after surgery.[4 6 9 15] Two cohort studies of patients aged 66 or older conducted in the USA and Canada found new postoperative prolonged opioid use at about 6% and 3%, respectively.[4 9] Sun et al[15] studied patients between the age of 18 and 64 years and found prolonged opioid use at around 1%. The discrepancy in occurrence of postoperative prolonged opioid use between these studies can at least partially be explained by differences in definitions of prolonged opioid use and how long a patient must be opioid naive for. Sun et al[15] used the most conservative definition of prolonged use where patients needed to fill 10 or more prescriptions or more than 120 days' supply of an opioid in the first year after surgery. While this definition captures the patients with a high postsurgical opioid consumption, it may underestimate the risk for patients with a lower, yet problematic, use. Although some patients will need opioids to manage postoperative acute pain, this period should be limited. Previous studies have shown that half of patients using opioids for longer than 90 days persist in their usage for many years after initiation.[16 17] This highlights the importance of restrictiveness in prescribing opioids. Two of these studies only included privately insured patients.[4 15] This introduces a bias in only selecting patients with the ability to afford a private insurance, potentially yielding a healthier population that is not generalisable to the actual population of the country. The lack of private insurances in a Swedish cohort is another strength of this study which adds additional insight to the field. Another aspect that could lead to differences in results between these studies is the lack of consensus regarding how to handle patients that died before the end of follow-up. We excluded these patients before analysis to avoid competing risk. In the Canadian cohort, patients that were unable to meet the primary outcome were also excluded[9] whereas the three other studies do not state this.[4 6 15]

## Patient-related risk factors for postoperative opioid use

The risk of prolonged opioid use was higher for women. This is not a consistent finding across the literature as other studies have found male sex[15] to be a greater risk factor, or no significant gender difference.[4 8 9] For comorbidities, a higher CCI score generated a higher aOR. We did not find a distinct trend regarding age where patients between 50 and 59 years of age had the highest aOR. Other studies have found both the oldest and the youngest population to be at higher risk, but also no significant difference in terms of age.[4 8 9 15] History of psychiatric disease and/or medication led to almost a twofold risk increase. Excluding the oldest population in a restriction analysis further increased the aOR for history of psychiatric disease, indicating that the impact of psychiatric disease as a risk factor for new postoperative prolonged opioid use is stronger in younger patients

(online supplemental table 4). In line with this, other studies have found an association between preoperative benzodiazepine or antidepressant use and prolonged postoperative opioid consumption.[10 18] Our results, alongside aforementioned studies, indicate that clinicians must be aware of this relationship when prescribing postoperative opioids.

### Surgery-related risk factors for postoperative opioid use

Acute surgery had a higher occurrence of postoperative opioid use compared with elective surgery. Being exposed to cancer surgery yielded an aOR of 1.06 (95% CI: 1.02 to 1.14). A patient going through cancer surgery is a potential confounder in our study as treating patients with cancer with opioids often is unavoidable. Partly, we tried to account for this by excluding patients prescribed opioids up to 180 days before surgery. Moreover, since individuals dying up to a year after surgery were excluded, we were hopefully able to account for unsuccessful surgical removal of the tumour and deterioration of advanced stages of malignancies. Our finding of a very low risk increase for cancer surgery speaks against this being a significant confounder.

### CONCLUSION

In this large national observational cohort study, 7% developed new prolonged opioid use postoperatively. We identified several potential risk factors where a history of psychiatric disease was the most important. Data on these susceptible patients could help clinicians reduce the number of prolonged opioid users by adapting their analgesic and preventative strategies. Clinicians need be aware of the risks, have a plan for tapering and realise that every opioid prescription has the potential of turning a non-user into a long-term user.

**Contributors** FL, MB and LH were responsible for the study concept and design. LH did the data collection and data cleaning. FL and LH did the data analysis. FL drafted the manuscript. MB, EL and LH did critical review of the manuscript. All the authors interpreted the data and approved the final manuscript as submitted and agree to be accountable for all aspects of the work. FL was responsible for the overall content as the guarantor.

**Funding** The authors have not declared a specific grant for this research from any funding agency in the public, commercial or not-for-profit sectors.

**Competing interests** None declared.

**Patient and public involvement** Patients and/or the public were not involved in the design, or conduct, or reporting, or dissemination plans of this research.

**Patient consent for publication** Not applicable.

**Ethics approval** The study protocol was approved by the Regional Ethics Committee of Stockholm, Sweden (2014/1306-31/3).

**Provenance and peer review** Not commissioned; externally peer reviewed.

**Data availability statement** Data may be obtained from a third party and are not publicly available. Data from Swedish health registers are available to all researchers upon making an application.

**ORCID iD**
Felix C B Lindeberg http://orcid.org/0000-0002-9462-4847

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
