## [Reviewer comments · BMJ Open]

ARTICLE DETAILS

TITLE (PROVISIONAL)	New prolonged opioid consumption after major surgery in Sweden: a population-based retrospective cohort study
AUTHORS	Lindeberg, Felix; Bell, Max; Larsson, Emma; Hallqvist, Linn

VERSION 1 – REVIEW

REVIEWER	Bicket, Mark C. University of Michigan
REVIEW RETURNED	23-Jan-2023

GENERAL COMMENTS	The authors present an observational cohort study of persons in Sweden undergoing major surgical procedures to understand the rate of opioid prescription fills in the period of time from 181 to 365 days after the surgical procedure, as well as risk factors for fills during that same time period. The study is innovative in terms of the data providing an international examination of opioid use after surgery, which has been examined in only few other countries outside of North American (e.g., Korea), and as such the authors would do well to emphasize unique features and insights provided by this new patient cohort. While the authors allude to differences in the definition, and have some flexibility in defining NPOU, the currently used definition leaves open the possibility that opioid use after surgery occurs for non-surgical means, which is a major limitation of the current definition and likely accounts for the strong association of age with NPOU. Major: The concept of new prolonged opioid use has evolved over time. This analysis uses a fairly liberal definition, given several reactions for opioid use 6 to 12 months after surgery, including greater likelihood of other diseases not related to surgery such as pain diagnoses, resulting in legitimate needs for opioids. To mitigate confounding, different approaches have been taken. One is to use a more strict definition that requires the fill of an opioid prescription in the immediate postoperative period of 0-3 days after discharge from surgery, along with fills over days 4-90 and 91-180. The authors also discuss other definitions in parts of the discussion. Without a sequence of early fills, or more restrictive criteria, it seems less likely that opioid use in later months actually represents opioid use that is appropriately attributed to surgery. Alternatively, the use of a non-surgical control group would provide some estimation of opioid fill rates that occur due to the natural progression of other diseases not related to surgery.
---

	Opioid naive definition - if data is available for 5 years, why limit the opioid naive period to 90 days? This seems less precise than longer intervals of being opioid naive of 6 months, a year, or more. Confounders - Table 1 shows several differences between the two cohorts. Adjusted odds ratios in the final model only adjust for a portion of those variables. As such, unaddressed confounding is a significant concern with respect to biases in the estimates provided in the analysis. Strengths and Limitations section: please clarify as lack of “longer” term follow up, given 12 months in many surgical cohorts may be considered long by some communities; the variation in definition and weakness in our study may be addressed in part per the discussion above. Other: Title: Please include the type of surgical procedures (“major” or “hospitalized” or otherwise) in the title as the cohort is not representative of all surgical procedures Abstract: Specifying some of the most prominent risk factors with estimates of the association (and precision) would enhance the abstract Introduction Overall the logic and flow are reasonable. The discussion about risk factors and hypothesis is generalized and could be used to add additional context or the study. Please consider alluding to the contribution that the Swedish cohort provides. What additional advantage comes from this group? Methods Page 8 line 13 - What characteristics for the study cohort would be anticipated to differ from the other 60% that is not captured in the study cohort? Please specify the date of data that was used for the study (the date for patient inclusion is listed but this differs from date of data analyzed) How is data quality from Orbit addressed? What identifiers were used for data linking? If data on five years before surgery was available, why only require 90 days of opioid free period before surgery? Why not use 1 year, given persons with intermittent fills have been shown to have higher rates of NPOU? Please include codes for classifying the subgroups in supplemental material Please enumerate the comorbidities in the methods that were examined, and their definitions. What is the time period of preoperative medications (5 years, 90 days, other?)? What are the types of surgeries, such as “acute” surgery? Are “Opioids with antispasmodics” certain types of opioids, or opioids prescribed with another type of medication? What is the reason for the sensitivity analyses? Their rationale is unclear, such as age, interaction examination, etc. Discussion
--	--

	The finding of age with NPOU likely has more to do with the higher incidence of chronic pain (i.e., arthritis), as well as comorbidities that preclude prescriptions for non-opioid analgesics such as NSAIDs and acetaminophen Other: Please rephrase “pain killing” given its imprecision and contribution to overprescribing in surgery and other fields of medicine Change “new and prolonged opioid use” to “new prolonged opioid use” Page 9 line 15 - unclear “75th percentile.” Page 9 line 31 - Remove “not appropriate” as it is certainly appropriate to involve patients in research. The discussion would be enhanced by describing how well the Orbit data represents all of Sweden, i.e., the other 60% of cases not captured by it. Figure 1 - Exclusion criteria here do not align with those reported in the manuscript. Please clarify what ASA-class mixing means Exclusion for time includes those with <30 days follow up and those who died. Thus it appears possible that persons may have variable follow up time of 31 to 365 days after surgery. Was this the case? If so, the logistic regression model would not be the appropriate statistical model for this cohort.
--	---

REVIEWER	Borys, Michał Medical University of Lublin, II Department of Anaesthesiology and Intensive Care
REVIEW RETURNED	13-Feb-2023

GENERAL COMMENTS	This a well-written manuscript concerning a relevant problem. Of course, it is a retrospective study, and thus could be biased. The main problem is that similar studies have been published.
---

VERSION 1 – AUTHOR RESPONSE

Reviewer: 1
Dr. Mark C. Bicket, University of Michigan
Comments to the Author:
Bmjopen-2022-071135

The authors present an observational cohort study of persons in Sweden undergoing major surgical procedures to understand the rate of opioid prescription fills in the period of time from 181 to 365 days after the surgical procedure, as well as risk factors for fills during that same time period. The study is innovative in terms of the data providing an international examination of opioid use after surgery, which has been examined in only few other countries outside of North American (e.g., Korea), and as such the authors would do well to emphasize unique features and insights provided by this new patient cohort. While the authors allude to differences in the definition, and have some flexibility in defining NPOU, the currently used definition leaves open the possibility that opioid use after surgery occurs for non-surgical means, which is a major limitation of the current definition and likely accounts for the strong association of age with NPOU.

Many thanks for your thorough review of our manuscript. Your comments and suggestions are valid, and important, and we believe this revised manuscript is significantly improved. We have carefully addressed all comments and modified the manuscript accordingly.

Major:

The concept of new prolonged opioid use has evolved over time. This analysis uses a fairly liberal definition, given several reactions for opioid use 6 to 12 months after surgery, including greater likelihood of other diseases not related to surgery such as pain diagnoses, resulting in legitimate needs for opioids. To mitigate confounding, different approaches have been taken. One is to use a more strict definition that requires the fill of an opioid prescription in the immediate postoperative period of 0-3 days after discharge from surgery, along with fills over days 4-90 and 91-180. The authors also discuss other definitions in parts of the discussion. Without a sequence of early fills, or more restrictive criteria, it seems less likely that opioid use in later months actually represents opioid use that is appropriately attributed to surgery. Alternatively, the use of a non-surgical control group would provide some estimation of opioid fill rates that occur due to the natural progression of other diseases not related to surgery.

We fully agree that the definition of prolonged opioid use is crucial. As mentioned, several suggestions have been used in the literature of which none is ideal, and you can argue back and forth how liberal the definition should be and what captures the problem most accurate.

In a previous study (Edwards et al. Risk factors for new chronic opioid use after hip fracture surgery *BMJ Open*. Mar 8 2021) the authors actually chose not to include prescription in the first quarter because it was considered to be associated with acute postoperative pain management. In another study (Ladha KS et al. Opioid Prescribing After Surgery in the United States, Canada, and Sweden. *JAMA Netw Open*. Sep 4 2019) a striking pattern was presented where it was observed that patients in Sweden were reluctant to dispense opioid prescriptions up to seven days after surgery compared to USA and Canada. Therefore, we do not think it should be obligate to dispense an opioid prescription in the first 30 days after surgery. Patients could spend some, or a majority, of the first 30 days after surgery still admitted to the hospital or in rehab. In Sweden, it is also common that patients leave the hospitals with a handful of pills instead of getting a prescription. These are factors that make us believe collecting an opioid prescription in the first couple of days after surgery should not be obligate to be classified as a prolonged user.

But we have discussed your major concern within the research group and agreed that your suggestion, with a stricter definition, would increase the likelihood that the opioid use may be attributed to surgery. Therefore, our new definition of NPOU require an opioid fill during the first 90 days, day 91-180, and during day 181-365.

To highlight this major limitation of our study and others in the field, we have also reorganized the limitations section and moved this issue to the top. Being the major limitations, it deserves to be highlighted first.

Opioid naïve definition - if data is available for 5 years, why limit the opioid naïve period to 90 days? This seems less precise than longer intervals of being opioid naïve of 6 months, a year, or more.

This is also a very important point, and we are glad to have the opportunity to elaborate on this. In Sweden, opioids are rarely prescribed for longer than 30 days, usually shorter periods such as a 2-week consumption, but 90-days prescriptions may exist. We are therefore fairly convinced that we are excluding patients with an active opioid use, when using a shorter window as the opioid naïve period. However, after another review of the literature and your valuable input we have decided to extend the opioid naïve period to 180 days. A longer period is possible, the data is available, but we believe we then may exclude many patients at risk of chronic use who are not yet there, and where the surgery may act as a trigger. Our aim is not to exclude a patient who had one opioid fill 3,5 years ago and after that has been opioid free but instead to exclude patients where the surgical procedure is not the trigger for future consumption. The literature includes several examples of studies using shorter durations. Ladha *et al.*, used 90 days, Edwards *et al.*, used 180 days and so did von Oelreich.

Edwards NM, et al. Risk factors for new chronic opioid use after hip fracture surgery: a Danish nationwide cohort study from 2005 to 2016 using the Danish multidisciplinary hip fracture registry. *BMJ Open*. 2021.

Ladha KS, et al. Opioid Prescribing After Surgery in the United States, Canada, and Sweden. *JAMA Netw Open*. 2019.

Von Oelreich E, et al. Risk factors and outcomes of chronic opioid use following trauma. *Br J Surg*. 2020.

Confounders - Table 1 shows several differences between the two cohorts. Adjusted odds ratios in the final model only adjust for a portion of those variables. As such, unaddressed confounding is a significant concern with respect to biases in the estimates provided in the analysis.

Thank you for highlighting this. We realize we have failed to describe this correctly in the manuscript.

Confounding is important and it has been addressed in the multivariable regression analysis. The adjusted odds ratios presented are mutually adjusted for variables with univariate significance, and considered clinically relevant, in the multivariate analysis age, sex, **comorbidities** (charlson comorbidity index), surgical procedure (including acute/cancer surgery) and year of surgery. This is detailed in the figure 2 heading, but for some reason it was not specified correctly in the manuscript. The initially considered risk factors are also detailed in supplementary table 2.

We have now rewritten the method section on p 8, 1st paragraph, with an accurate description of the management of confounders.

Strengths and Limitations section: please clarify as lack of “longer” term follow up, given 12 months in

many surgical cohorts may be considered long by some communities; the variation in definition and weakness in our study may be addressed in part per the discussion above.

We agree that the formulation was weak from our side and have tried to clarify.

Other:

Title: Please include the type of surgical procedures (“major” or “hospitalized” or otherwise) in the title as the cohort is not representative of all surgical procedures

The title has been updated as suggested.

Abstract: Specifying some of the most prominent risk factors with estimates of the association (and precision) would enhance the abstract

Great suggestion. Estimates of the most important risk factors are included in the result section of the abstract.

Introduction

Overall the logic and flow are reasonable. The discussion about risk factors and hypothesis is generalized and could be used to add additional context or the study.

Please consider alluding to the contribution that the Swedish cohort provides. What additional advantage comes from this group?

True. The importance of studying postoperative opioid consumption in a Swedish setting, and compared to other countries, is now added in the introduction, p 3, 3rd paragraph, 2nd line.

Methods

Page 8 line 13 - What characteristics for the study cohort would be anticipated to differ from the other 60% that is not captured in the study cohort?

Good question. The data is extracted from 23 Swedish hospitals at university-, county-, and district level, covering approximately 40 % of Sweden, and is a representative cohort of all Swedish perioperative care. Since the included hospitals are located in different legislative regions and dispersed geographically, we have no reason to believe that the external validity would be low. This was mentioned in Material and Methods but is now described in more detail, p 5, 1st paragraph, line 3.

Please specify the date of data that was used for the study (the date for patient inclusion is listed but this differs from date of data analyzed)

This information is now added on p 9, last paragraph, first line.

How is data quality from Orbit addressed?

This is an excellent question. The orbit surgical planning system is reviewed, and the quality is assured. The hospitals using Orbit have 100% coverage of performed surgical procedures since it is

impossible to perform surgery without admitting the patient through orbit. Orbit surgical planning system is used in other countries, e.g., Denmark, Germany, and the Netherlands.

Moreover, the information in Orbit has been validated in three publications, previously conducted by two of the authors in this study (Hallqvist, Bell). In these studies, data on 300, 460 and 652 patients was extracted from Orbit and thereafter further information was retrieved from electronic medical records and anaesthetic charts. The information in Orbit, such as date, time and type of surgery and anaesthesia, was 100% correct.

1. Intraoperative hypotension and MI-development among high-risk patients undergoing non-cardiac surgery: A nested case-control study
Hallqvist L, Granath F, Fored M, Bell M.
Anesth Analg. 2021 Jul 1;133(1):6-15

2. Intraoperative hypotension is associated with acute kidney injury in noncardiac surgery: An observational study.
Hallqvist L, Mårtensson J, Granath F, Sahlén A, Bell M.
Eur J Anaesthesiol. 2018 Jun;33(6):450-6.

3. Intraoperative hypotension is associated with myocardial damage in noncardiac surgery: An observational study.
Hallqvist L, Granath F, Huldt E, Bell M.
Eur J Anaesthesiol. 2016 Apr;35(4):273-279.

What identifiers were used for data linking?

Data were linked using the Swedish personal identification number (PIN) of the participants found through the participant identification in Orbit, detailed on p 6, 1st paragraph. The PIN has been reviewed in Ludvigsson JF, *et. al.* The Swedish personal identity number: possibilities and pitfalls in healthcare and medical research. *Eur J Epidemiol.* 2009;24(11):659-67. doi:10.1007/s10654-009-9350-y.

If data on five years before surgery was available, why only require 90 days of opioid free period before surgery? Why not use 1 year, given persons with intermittent fills have been shown to have higher rates of NPOU?

Yes, this is a valid point. This issue is discussed above in an earlier segment.

Please include codes for classifying the subgroups in supplemental material

Please see the added supplementary table 4 with the surgical codes.

Please enumerate the comorbidities in the methods that were examined, and their definitions. What is the time period of preoperative medications (5 years, 90 days, other?)? What are the types of surgeries, such as “acute” surgery? Are “Opioids with antispasmodics” certain types of opioids, or opioids prescribed with another type of medication?

Thank you for bringing up this issue. Obviously, we should have included this in the first place. In the Table footnotes we have enumerated the comorbidities, what time period was used for preoperative medication and defined acute surgery. Regarding your question about opioids with antispasmodics that is a type of medication where an opioid is combined with an antispasmodic. We have added an example of the most common medication name to Table 1 for more clarity.

What is the reason for the sensitivity analyses? Their rationale is unclear, such as age, interaction examination, etc.

True. When using the stricter definition of prolonged opioid use, higher age is no longer a strong risk factor, and we have removed the interaction analyses between age and history of psychiatric disease.

The rationale for the restriction analysis is now described in more detail in p 8, 1st paragraph, line 7-10. We were interested in the impact of history of psychiatric disease in younger patients.

Discussion

The finding of age with NPOU likely has more to do with the higher incidence of chronic pain (i.e., arthritis), as well as comorbidities that preclude prescriptions for non-opioid analgesics such as NSAIDs and acetaminophen

Thank you for your input. After changing the definition of prolonged use and number of days without opioids before surgery we did not see the same trend regarding higher age. Therefore, this part of the discussion has been subject to change.

Other:

Please rephrase "pain killing" given its imprecision and contribution to overprescribing in surgery and other fields of medicine

Very good point. We have rephrased.

Change "new and prolonged opioid use" to "new prolonged opioid use"

Thank you for this suggestion. It has been updated.

Page 9 line 15 - unclear "75th percentile."

Thank you for this excellent comment. We agree that this was not very clear. We have now excluded the sensitivity analysis where this was mentioned.

Page 9 line 31 - Remove "not appropriate" as it is certainly appropriate to involve patients in research.

Correct, thank you!

The discussion would be enhanced by describing how well the Orbit data represents all of Sweden, i.e., the other 60% of cases not captured by it.

We agree and have now included a discussion about this on page 5, 1st paragraph, line 3-5.

Figure 1 -

Exclusion criteria here do not align with those reported in the manuscript.

Please clarify what ASA-class mixing means

Exclusion for time includes those with <30 days follow up and those who died. Thus it appears

possible that persons may have variable follow up time of 31 to 365 days after surgery. Was this the case? If so, the logistic regression model would not be the appropriate statistical model for this cohort.

Thank you for these observations. Regarding the ASA-missing question: The cohort was refined by excluding patients from hospitals with a high percentage of missing ASA-classification. After exclusion, the percentage of patients with missing ASA was below 10%: hence no imputation of data was considered needed. Exclusions had no impact on crude or adjusted relative risks, analyses indicated ASA-classification were missing completely at random. Regarding the follow up all patients have the same follow up time from index, i.e. time of surgery. Hence, there no risk of differences in follow up time.

Reviewer: 2

Dr. Michał Borys, Medical University of Lublin

Comments to the Author:

This a well-written manuscript concerning a relevant problem.
Of course, it is a retrospective study, and thus could be biased.
The main problem is that similar studies have been published.

Thank you for your input. Although similar studies have been published most of them cover North American populations. Therefore, we think a Swedish studies would add to the field since Swedish prescribing patterns differ from North American populations in the days following surgery. Additionally, the lack of private insurances in the Swedish medical setting is another difference from many North American cohorts, potentially offering a less biased population.

Reviewer: 1

Competing interests of Reviewer: No competing interests

Reviewer: 2

Competing interests of Reviewer: None